# Cycloastragenol Inhibits Experimental Abdominal Aortic Aneurysm Progression

**DOI:** 10.3390/biomedicines10020359

**Published:** 2022-02-02

**Authors:** Leander Gaarde Melin, Julie Husted Dall, Jes S. Lindholt, Lasse B. Steffensen, Hans Christian Beck, Sophie L. Elkrog, Pernille D. Clausen, Lars Melholt Rasmussen, Jane Stubbe

**Affiliations:** 1Centre for Individualized Medicine in Arterial Diseases (CIMA), Odense University Hospital (OUH), 5000 Odense, Denmark; leander.gaarde@rsyd.dk (L.G.M.); julie.husted.dall@regionh.dk (J.H.D.); jes.sanddal.lindholt@rsyd.dk (J.S.L.); hans.christian.beck@rsyd.dk (H.C.B.); lars.melholt.rasmussen@rsyd.dk (L.M.R.); 2Department of Cardiothoracic and Vascular Surgery, Odense University Hospital, 5000 Odense, Denmark; 3Cardiovascular and Renal Research Unit, Institute for Molecular Medicine, University of Southern Denmark, 5000 Odense, Denmark; lsteffensen@health.sdu.dk (L.B.S.); soelk17@student.sdu.dk (S.L.E.); pecla17@student.sdu.dk (P.D.C.); 4Department of Clinical Biochemistry and Pharmacology, Odense University Hospital, 5000 Odense, Denmark

**Keywords:** aortic aneurysm, pathogenesis, pharmacological therapy, experimental model, drug delivery

## Abstract

The pathogenesis of abdominal aortic aneurysm involves vascular inflammation and elastin degradation. *Astragalus*
*radix* contains cycloastragenol, which is known to be anti-inflammatory and to protect against elastin degradation. We hypothesized that cycloastragenol supplementation inhibits abdominal aortic aneurysm progression. Abdominal aortic aneurysm was induced in male rats by intraluminal elastase infusion in the infrarenal aorta and treated daily with cycloastragenol (125 mg/kg/day). Aortic expansion was followed weekly by ultrasound for 28 days. Changes in aneurysmal wall composition were analyzed by mRNA levels, histology, zymography and explorative proteomic analyses. At day 28, mean aneurysm diameter was 37% lower in the cycloastragenol group (*p* < 0.0001). In aneurysm cross sections, elastin content was insignificantly higher in the cycloastragenol group (10.5% ± 5.9% vs. 19.9% ± 16.8%, *p* = 0.20), with more preserved elastin lamellae structures (*p* = 0.0003) and without microcalcifications. Aneurysmal matrix metalloprotease-2 activity was reduced by the treatment (*p* = 0.022). Messenger RNA levels of inflammatory- and anti-oxidative markers did not differ between groups. Explorative proteomic analysis showed no difference in protein levels when adjusting for multiple testing. Among proteins displaying nominal regulation were fibulin-5 (*p* = 0.02), aquaporin-1 (*p* = 0.02) and prostacyclin synthase (*p* = 0.007). Cycloastragenol inhibits experimental abdominal aortic aneurysm progression. The suggested underlying mechanisms involve decreased matrix metalloprotease-2 activity and preservation of elastin and reduced calcification, thus, cycloastragenol could be considered for trial in abdominal aortic aneurysm patients.

## 1. Introduction

Abdominal aortic aneurysm (AAA) is a localized enlargement of the aorta exceeding 3 cm in diameter and is potentially life-threatening [1]. Globally, AAA rupture is a major cause of mortality in elderly men, being responsible for the death of 1% of men above 65 years [1,2]. Today, patients with known AAAs are carefully monitored and offered surgical repair when the AAA possesses a diameter above 5–6 cm. Thus, an urgent unmet clinical need of medical therapies for small AAAs exists, to prevent progressive dilatation, acute or elective surgical repair, rupture, and death [3,4,5].

Chronic inflammation, due to persistent infiltration of inflammatory cells into the aortic wall and degradation of elastin, seems to fundamentally characterize the pathology of AAAs [6]. Elastin in the medial layer of the aortic wall is degraded when the aneurysm is formed. This loss of elastin is partly compensated by the continuous formation of collagen and elastin by vascular smooth muscle cells along with expression of structural proteins such as tropoelastin and fibrillin 1, which is cross-linked by lysyl oxidase (LOX) and fibulin-5 to form elastic fibers [6,7,8]. This process is diminished by inflammation when elastin-degrading enzymes, such as matrix metalloproteases (MMPs), are released from macrophages, monocytes, neutrophils and activated vascular smooth muscle cells [6,7]. Especially the M1 proinflammatory macrophages, which produce proinflammatory cytokines and MMPs, augment AAA progression [9]. Moreover, reactive oxygen species (ROS) contribute to AAA expansion by enhancing MMP activity, which further degrades the extracellular matrix (ECM) and weakens the aneurysm wall [6,10,11,12,13].

A promising dietary supplement in the fight against AAA progression is cycloastragenol (CAG). It is a crystalline solid triterpenoid saponin compound and a hydrolyzed product of the main active ingredient in the Chinese herb *Astragalus membranaceus*. CAG has been used in traditional Chinese medicine for over 2000 years with no commonly known side effects [14]. Recent literature describes various anti-inflammatory effects of CAG in heart, vascular, liver and skin tissue including inhibition of lymphocyte activation. CAG also reduces MMP-2 and MMP-9 expression and activity, thereby preserving extracellular matrix (ECM) integrity [15,16,17,18,19]. CAG supplement has recently been proven to attenuate AAA expansion in mice using an elastase wrapping model and in angiotensin-II induced AAA in Apolipoprotein E (ApoE)^−/−^ mice [19]. However, the model can be criticized for not producing human-like-AAA in contrast to the elastase perfusion model in rats, which uses porcine pancreatic elastase (PPE) applied intraluminally in the infrarenal aortic segment through laparotomy and atherectomy [20]. This imitates human AAA well, as it displays various pathological similarities, such as inflammation, elastin degradation, thrombus formation and calcification [21]. Thus, the growing base of evidence describing the anti-inflammatory properties of CAG illustrates its potential as a possible future medical treatment against AAA expansion in humans.

In this study, we hypothesize that CAG inhibits the progressive dilatation of AAA in the rat PPE aneurysm-model by its anti-inflammatory and anti-oxidative effects leading to reduced protease activity and, thereby, preserving elastin integrity.

## 2. Materials and Methods

### 2.1. Study Design

The rats were randomly allocated to either CAG treatment (125 mg/kg/day) or controls, starting on the first postoperative day, by an external investigator, not participating in any of the experimental procedures. Therefore, rats were housed independent of treatment, and daily caretakers were blinded to treatment.

### 2.2. Outcomes

The primary outcome was peak-systolic infrarenal aortic anterior to posterior inner-to-inner diameter. Secondary explanatory outcome were: AAA wall content of elastin, its structure and LOX mRNA levels. Determine the effect of CAG on matrix-dependent MMP-2, -9 and -12 mRNA levels and MMP activity. Measure mRNA levels of leukocyte marker CD45 mRNA, macrophage marker F4/80 mRNA and IL-6 and -10 mRNA levels together with the pro-inflammatory M1 macrophage marker iNOS, the antioxidative markers Nrf2 and HO-1 in the aneurysms. Furthermore, perform histological assessments of cluster of differentiation (CD)68 and CD206 and the presence of calcification in the aneurysm wall. Finally, identify potential new targets of CAG treatment in aneurysm wall by discovery proteomic analysis.

Potential harm outcomes were weight of liver, spleen, heart, and kidneys, as well as morphology of the inferior right liver lobe.

### 2.3. Sample Size Calculation

We have, in previous experiments with the rat model, observed a mean diameter increase of 158.75% ± 77.5 SD. To detect a 50% difference, which is considered clinically relevant, by a *t*-test using 5% significance level and 80% power, 24 rats are needed (12 in each group). This sample size estimation is conservative, as two-way repeated measure ANOVA tests were used to determine the correlation of the aortic diameters between the groups over time.

### 2.4. Experimental Animals, Ethical Statement, Housing and Husbandry

Male Sprague-Dawley rats purchased from Janvier Laboratories, Le Genest-Saint-Isle, France, were housed in cages of up to 4 rats per cage under twelve-hours light/dark cycle, room temperature of 20 °C, air humidity of 55% with free access to standard chow and tap water at the Biomedical Laboratory at the University of Southern Denmark. Rats were acclimatized for at least one week after delivery before entering the experimental protocol.

All animal experiments were conducted in accordance with a protocol ethically approved by the Danish Animal Experiments Inspectorate (license nr. 2016−15−0201−01046), and in accordance with arrive guidelines [22]. As females are generally protected against AAA formation [1,2,23,24], we only used males in this study.

### 2.5. Induction of Abdominal Aortic Aneurysm by Perfusion of Pancreatic Porcine Elastase (PPE) in the Infrarenal Region of the Aorta

On the day of surgery, male Sprague-Dawley rats (260–435 g corresponding to age 7–10 weeks) were given 0.2 mg temgesic (buprenorphine, Indivior, North Chesterfield, VA, USA) administered in 1 g of nut paste (Nutella) for pain management [25]. Then, the rats were anesthetized by a subcutaneous injection of a mixture of fentanyl (236 µg/kg), fluanisone (7.5 mg/kg, Skanderborg Apotek, Skanderborg, Denmark) and midazolam (3.75 mg/kg, Hameln Pharma, Hamelin, Germany) and underwent AAA induction by intraluminal PPE infusion of the infrarenal region by the procedure previously described by Shack et al. [26]. The only variation was that pancreatic porcine elastase concentration was increased to 12 units/mL for 30 min and post-surgical pain management was provided with additional 0.2 mg temgesic in nut paste (Nutella) [25]. The surgical procedure lasted between 60–70 min. If needed rats were supplemented with 20% of the initial dose of anesthesia by subcutaneous injection during the operation. Representative pictures of the surgical steps and post-surgery day 28 are shown in Figure 1. All rats were observed post-surgery until they were fully awake from anesthesia. They were housed in individual cages in a heated cabinet until next morning, where treatment of the rats was initiated, and the rats were housed together throughout the rest of the experimental period. Rats were treated daily from day 1 post-surgery with CAG (125 mg/kg, Chengdu King-tiger Pharm-chem. Tech. Co., Ltd., Chengdu, China, n = 12) or vehicle (deionized reverse osmosis water containing 0.05% methylcellulose (*w*/*v*, Merck), 2% tween 80 (*v*/*v*, Merck), n = 12) by oral gavage using Soft Flex feeding tubes (Vetagree, Oslo, Norway) for 28 consecutive days; AAA expansion was monitored weekly by ultrasound measurements, as described below. Four rats were excluded; two vehicle controls died due to surgical and post-surgical complications and two did not consistently receive CAG due to difficulties with oral gavage.

### 2.6. Ultrasound Measurement of Aneurysm Progression

The abdominal aorta was video recorded from the renal artery to the bifurcation using ultrasound (LogiQ e ultrasound machine and a L10-22-RS transducer, GE Healthcare, Brøndby, Denmark) on the day of AAA induction, day 0 prior to surgery and, thereafter, on days 7, 14, 21 and 28 during treatment using 4% isoflurane inhalation anesthesia (Sigma-Aldrich, Søborg, Denmark). All ultrasound scans were performed by the same investigator, as our pilot study of 76 ultrasound recordings from 16 rats displayed a 6.9% variance between two investigators and stored for later analysis. Measurements of maximal vertical anterior to posterior diameter of the aorta spanning from internal edges during peak systolic blood pressure were performed blinded to the treatment group by two independent investigators using standard software on the LogiQ e Ultrasound machine. Inter-observer variation of the measured diameters was determined to be 3.7% based on the 76 pilot study ultrasound recordings. Values for the relative increase were obtained by adjusting to the diameter on day 0.

### 2.7. At Termination

All rats were euthanized by exsanguination 28 days after AAA induction, resulting in 10 rats in each group. Liver, spleen, heart, and kidneys were collected and weighed. Subsequently, a specimen from the inferior right liver lobe was fixed as described below for morphological analysis.

The aneurysm specimens were isolated and divided into two pieces; the cranial piece was fixed in a 10% normal formalin buffer (Hounisen Laboratorie udstyr A/S, Skanderborg, Denmark) over night at 4 °C, then placed in phosphate-buffered saline (PBS) (Thermo Fisher, Slangerup, Denmark) with 0.05% azide (Sigma-Adrich, Søborg, Denmark) and subsequently embedded in paraffin for morphological analysis. Two samples from the vehicle-group and three from the CAG-group, were unfortunately damaged in the embedding process. The caudal piece was immediately snap frozen in liquid nitrogen and kept at −80 °C until RNA and protein isolation.

### 2.8. Miller’s Elastin and Calcium Von Kossa’s Staining

Five µm cross-sections of paraffin embedded aneurysm specimens were sectioned, deparaffinized and hydrated. For identification of elastin fibers, Miller’s elastin stain kit (Atom Scientific, Hyde, UK) was used according to the manufacturer’s instructions. In brief, sections were stained for 3 h in Miller’s elastin stain, subsequently washed and counterstained with Van Gieson’s stain.

Micro-calcium deposits in the aneurysm wall were detected by the silverplating kit according to Von Kossa’s stain instructions (Merck, Søborg, Denmark). Calcium deposits were visualized using a 20-watt energy-saving lamp (Quantification method, described below). One additional sample from the vehicle group was damaged during the Von Kossa staining process.

### 2.9. Immunohistochemistry

Aneurysm cross-sections were deparaffinized and hydrated, followed by antigen retrieval by heating to 100 °C for 15 min in a citrate buffer (10 mM; pH 6, Merck, Søborg, Denmark) for CD206, MMP2or in a TEG-buffer (10 mM TrisBase; 0.5 mM EGTA; pH 9, Sigma Aldrich, Søborg, Denmark) for CD68.

Sections were subsequently blocked for endogenous peroxide activity using, respectively, a 3% and a 1.5% hydrogen peroxide (Merck, Søborg, Denmark) in tris-buffered saline (TBS, Sigma Aldrich, Søborg, Denmark) solution for, respectively, 10 and 30 min. This was followed by one hour blocking in a 5% milk/1× TBS/0.05% Tween-20 (TBST) solution for CD68 and α-actin 3% BSA-TBST solution for CD206 and MMP2.

After washing in TBST, the aneurysm sections were incubated overnight at 4 °C with primary anti-CD68 (Abcam, Cambridge, UK) 1:500 and α-actin (Sigma Aldrich, Søborg, Denmark) 1:500 in 5% milk/TBST and anti-CD206 (Abcam, Cambridge, UK) 1:1000 and MMP2 (Abcam, Cambridge, UK) 1:500, both in 3% BSA/TBST. The next day sections were washed in TBST and incubated with horseradish peroxidase (HRP) conjugated goat-anti-rabbit (DAKO, Glostrup, Denmark) 1:1000 or HRP conjugated anti mouse (DAKO, Glostrup, Denmark) 1:1000 in 5% milk/TBST or 3% BSA/TBST. Positive staining was visualized with 3,3′-Diaminobenzidine tetrahydrochloride hydrate (DAB, Merck, Søborg, Denmark) and sections were counterstained in Mayer’s hematoxylin (Merck, Søborg, Denmark) and rinsed in tap water. As negative controls rabbit immunoglobulin IgG corresponding to the primary antibody concentration was applied (DAKO, Glostrup, Denmark). All staining was analyzed in an Olympus Bx51 microscope and micrographs were captured using an Olympus DP26 camera. After analysis, whole frame micrographs were adjusted for brightness and contrast using Photoshop (ver. 9, San Jose, CA, USA).

### 2.10. Elastin Content Analysis and Immunohistochemical Cell Count

For the assessment of elastin percentage in the medial layer, Image J software (ImageJ 1.53a Wayne Rasband, National Institutes of Health, Bethesda, MD, USA) was used. The external edge of the medial layer was defined as the transition site from disrupted or concentric rings of elastin to connective tissue in the adventitial layer. To quantify the percentage of elastin, the color threshold tool was used. For the scoring of aneurysmal wall disruption, each micrograph was divided into 8 areas, and each field was scored from 1–4, 4 being severe wall disruption and 1 minimal wall disruption. All assessments of elastin content were performed by two investigators blinded to treatment. The interobserver variation was 2.2% and the average score was used for statistical calculations.

Thereafter, elastin lamellae externa was used to identify the border between the medial layer and adventitial layer when identifying the adventitial area with CD68 and CD206 positive cell count per mm^2^. The total area of adventitial layer was divided by the number of positive labeled cells to determine positive cells per mm^2^. One investigator blinded to treatment determined numbers of positive CD68 and CD206 cells per adventitial area.

### 2.11. Zymography

Aneurysm samples were homogenized in protein extraction buffer (0.3 M sucrose; 25 mM Imidazole, 1 mM EDTA, pH 7.2 complete protease inhibitor cocktail 2 and 3, Sigma Aldrich, Søborg, Denmark). Samples were centrifuged for 10 min at 6000× *g* at 4 °C. Protein concentration was determined by Bicinchoninic Acid Kit for Protein Determination (Sigma Aldrich, Søborg, Denmark) using bovine serum albumin as the standard. A total of 12 µg protein samples and 1.25 µL recombinant MMP-2 (Sigma Aldrich, Søborg, Denmark) were mixed with an equal amount of 2× tris-glycine SDS sample buffer (Thermo Fischer) loaded onto a Novex zymogram gel containing 10% gelatin (Thermo Fisher, Slangerup, Denmark) and proteins were separated by gel electrophoresis at 125 V for 90 min. Proteins were then allowed to refold 30 min in renaturation buffer (Thermo Fisher, Slangerup, Denmark) followed by 24 h at 37 °C in developing buffer (Thermo Fischer, Slangerup, Denmark). Finally, undigested proteins in the gel were stained with simple blue stain (Thermo Fisher) for 30 min. White bands were inverted and quantified in Molecular Imager Image Lab (ChemiDoc WRS+, Biorad, Copenhagen, Denmark).

### 2.12. Proteomic Analysis

Preparation of AAA tissue for mass spectrometry was performed as previously described [27]. In brief, snap frozen tissue was homogenized in a lysis buffer, then denatured, alkylated, and digested with trypsin overnight. Tryptic peptides were purified on custom-made Poros R2/R3 (Thermo Fisher Scientific, Slangerup, Denmark) columns, and peptide concentration was normalized across samples. Samples (4 μg tryptic peptides per sample) were randomly labelled with 10-plex tandem mass tags (TMT, Thermo Scientific, Waltham, MA, USA); mass tag 126 was a pool of all AAA samples and served as internal control. Proteome data are protein abundances relative to the internal control. Mixed peptide samples were high-pH fractionated and analyzed by nano-LC–MS/MS virtually, as previously described [28]. All Eclipse raw data files were processed and quantified using Proteome Discoverer version 2.4 (Thermo Scientific, Waltham, MA, USA) as previously described [28].

### 2.13. Quantitative Polymerase Chain Reaction Measurements (qPCR)

The methods of total RNA isolation, cDNA synthesis and qPCR quantification have been described previously by Wintmo et al. [29]. Addition of 1 µL glycoblue (Thermo Fisher, Slangerup, Denmark) as a carrier for enhancement of RNA precipitation was the only modification. Primers used for determining mRNA levels are shown in Table 1.

Messenger RNA (mRNA) levels of genes of interest and five standards (10-fold dilutions) were run in duplicate using SYBR green (Biorad, Copenhagen, Denmark) as the detector system. RNase-free water and RNA samples without reverse transcriptase were used as negative controls. All samples were loaded on 96 Aria Max well plates (Agilent Technologies, Santa Clara, CA, USA), and the PCR amplification was done using three steps (initial 3 min at 95 °C, followed by 40 cycles: 95 °C 20 s; 60 °C 20 s, 72 °C 15 s) followed by a melting curve for the determination of PCR-product selectivity. RNA yield in one sample from each group was low; therefore, these samples were only included in RPL41, LOX, F4/80 and iNOS. Each mRNA expression level of the gene of interest was normalized to the complimentary expression level of the housekeeping gene ribosomal protein L41 (RPL41) that we first tested and did not change significantly between vehicle- and CAG-treated aneurysms.

### 2.14. Statistical Methods

D’Angostino and Pearson test was used for normality testing. A two-way repeated measures ANOVA adjusted for weight at entry with a Greenhouse-Geisser correction, due to the violation of the assumption of sphericity (Mauchly’s test), was used to analyze difference in relative aneurysm diameter between groups, calculated in SPSS (IBM SPSS Statistics, IBM Corporation, Endicott, NY, USA, 1989, 2020). Sidak’s multiple comparison test was subsequently applied in Graphpad Prism (ver. 8, San Diego, CA, USA) for each time point.

For the secondary explanatory data, two-tailed unpaired Student’s t-test was used to analyze normally distributed data. Values are presented as mean ± standard deviation (SD). Welch’s correction was used if F-test for variance was significant. A non-parametric Mann–Whitney test was used if data failed normality testing by the D’Angostino and Pearson test. Values are then presented as median ± interquartile range (IQR). Chi-square test was used for categorial outcome variables.

Explorative proteomics data were analyzed by Student’s t-test for each protein and subsequent false discovery rate (FDR) correction for multiple testing and GO enrichment analysis was performed using default settings of the DAVID Bioinformatics Resources [30,31]. The *p*-values < 0.05 were considered significant.

## 3. Results

All rats tolerated daily treatment well, except for the two CAG rats excluded due to difficulties with the daily gavage. Treatment did, however, seem to affect liver, spleen, and heart to body weight ratios with a higher ratio among CAG treated rats (Appendix A), while kidney to body weight ratio was unaffected (Appendix A). Microscopic analysis of HE-stained liver lobes displayed no obvious differences as evaluated by investigators. As mentioned, a total of four rats were excluded causing an unintendedly higher non-significant mean initial body weight in the vehicle treated rats when compared to the CAG-treated group. Furthermore, there was a large variation in body weight within groups (vehicle: 351.8 g ± 52.2 g vs. CAG: 332.0 g ± 50.1 g, *p* = 0.33, n = 10/10). To make sure initial body weight did not influence AAA expansion, the statistical analysis was adjusted for body weight. Body weight increase during the experimental period was similar in both groups (Appendix A, 35.6% ± 11.3% vs. 36.78% ± 11.2%; *p* = 0.83, n = 10/10).

### 3.1. CAG Treatment Inhibited AAA Expansion

The relative increase in aortic aneurysm diameter at the widest point during peak systolic blood pressure, adjusted for weight at entry, increased gradually in both groups during the experimental period of 28 days (Figure 2). CAG treatment led to significantly smaller aneurysms on days 7, 14, 21 and 28 compared to vehicle treatment (Figure 2). Aneurysm growth was most pronounced during the first 14 days after induction and reached maximal enlargement after 21 days with a mean relative increase of 124% ± 10% and 88% ± 10% for vehicle and CAG groups, respectively (Figure 2). No further change in aneurysm growth was observed at day 28 in either group.

### 3.2. CAG Treatment Affects Elastin Integrity

Comparing cross sections of aneurysms at day 28 from both groups to unaffected abdominal aortas proximal to the aneurysm, revealed significant degradation and disruption of elastin lamellae in the medial layer (Figure 3A). Assessing the elastin content in the medial layer in both the vehicle and CAG treated AAA sections showed the mean percentage of elastin content was nearly doubled in the CAG treated group compared to the vehicle treated group, though not significantly (Figure 3B, *p* = 0.20). Elastin degradation and disruption was not affected uniformly in the aneurysm cross sections. Scoring 8 areas on each cross-section showed that areas more prone to rupture (grade 4) were significantly more pronounced in the vehicle-treated group, while larger areas in the CAG-treated group were minimally affected, with large areas scoring grade 1 (Figure 3C, *p* = 0.0003), indicating less wall thinning and destruction, thus, less potential for rupture. This protective effect on elastin was most likely not caused by augmented synthesis of elastin, as the mRNA levels of lysyl oxidase (LOX), an enzyme involved in elastin synthesis and cross-linking [7], was similar in both groups (Figure 3D, *p* = 0.57).

To investigate whether CAG protects against elastin degradation, mRNA levels of MMPs known to play a major role in AAA development [32] were determined. Neither mRNA levels of MMP-2 (*p* = 0.22), MMP-9 (*p* = 0.24) nor MMP-12 (*p* = 0.60) were affected by CAG treatment (Figure 4A–C). In contrast, MMP-2 activity measured by zymography was significantly decreased in the CAG-treated group (Figure 4D), suggesting CAG treatment partly prevents elastin degradation and AAA growth by dampening MMP-2 activity. MMP-2 was associated with a subset of vascular smooth muscle cells in the aneurysm wall, where weak labeling of MMP-2 was detected in the medial layer. There was no apparent difference between the two groups (Figure 4E).

### 3.3. The Effect of CAG on Infiltration of Inflammatory Cells into the Aneurysm Wall

Next, we determined the suggested anti-inflammatory properties of CAG by determining infiltration of immune cells into the aneurysm wall. The mRNA levels of the common lymphocyte marker CD45 (Figure 5A, *p* = 0.63) and the monocyte/macrophage marker F4/80 (Figure 5B, *p* = 0.44) were not affected in the CAG treated aneurysms.

The number of infiltrating macrophages identified as CD68 positive cells, localized to the adventitial layer of the aneurysms, did not show any difference between groups (Figure 5C, D, *p* = 0.56).

As the balance between pro-inflammatory M1 macrophages and tissue repairing M2 macrophages has previously been shown to be important for AAA expansion [9], the M2 macrophages identified as CD206 positive cells were determined in the aneurysm wall (Figure 6A). CD206 positive cells were limited to the adventitial layer, and there were no differences in the number of CD206 positive cells per mm^2^ between the two groups (Figure 6B, *p* = 0.99). In agreement, there were no difference in the aneurysmal mRNA levels of the anti-inflammatory cytokine IL-10 between groups (Figure 6C, *p* = 0.114). Moreover, levels of iNOS mRNA, another marker for M1 macrophages, did not differ between the vehicle and CAG treated groups (Figure 6D, *p* = 0.684) and there was no difference in aneurysmal mRNA levels of the pro-inflammatory cytokine IL-6 between groups (Figure 6E, *p* = 0.340). Thus, the inflammatory response seemed not to be affected by CAG treatment.

### 3.4. The Effect of CAG on Oxidative Stress and Calcification of the Aneurysm Wall

To determine if CAG dampens AAA progression by reducing oxidative stress, the levels of antioxidative marker Nrf2 mRNA and its downstream target HO-1 were determined in the aneurysms. Neither Nrf2 nor HO-1 mRNA levels differed between treatment groups (Figure 7A,B; *p* = 0.171 and *p* = 0.489, respectively).

In more advanced AAAs, calcifications become significant [33]; therefore, the effect of CAG on calcifications was examined by Von Kossa’s calcium deposit staining. Calcifications were present in 4 out of 7 AAA samples in the vehicle treated group, while no calcifications were detected in 7 out of 7 in the CAG-treated group (Figure 8A,B, *p* = 0.018).

### 3.5. The Effect of CAG AAA Protein Composition Using Explorative Proteomics

To identify new mechanisms of CAG in limiting aneurysm progression, protein samples of the aneurysms were analyzed by liquid chromatography mass spectrometry (LC-MS/MS). We identified 2011 unique proteins (minimum n = 3/3), of which 57% were detected across all samples (n = 10/10) (Appendix A). No significant differences were found between CAG-treated and vehicle-treated aneurysms when correcting for multiple testing (Appendix A); thus, one should bear in mind that some unadjusted de-regulated proteins might be false positive.

The top 20 de-regulated proteins in the aneurysm wall are shown in Table 2. The proteins identified in the aneurysm wall include the structural fibulin-5 involved in elastin assembly [8], the anti-aggregatory and vasodilatory PGI_2_-producing enzyme prostacyclin synthase [34], and the water channel aquaporin-1 (AQP1). Table 2: Top 20 hits of proteins deregulated in CAG treated AAA tissue compared to vehicle treated AAA by explorative proteomics (n = 10/10). In this table, data is not adjusted for multiple testing. FC: fold change.

### 3.6. Effect of CAG on Vascular Smooth Muscle Cells

To determine if vascular smooth muscle cell layers in the aneurysm wall were changed by CAG treatment, the α-actin positive cells were examined in aneurysm cross-sections from the two groups. Intense α-actin labeling was detected in the medial layer of the aneurysms in both groups; there was no difference in the area of positive α-actin staining between vehicle and CAG-treated rats (Figure 9A,B, *p* = 0.56). That no major change in vascular smooth muscle cell layer was observed was further supported by the quantitative proteome analyses of the aneurysm wall showing no change in proteins associated with vascular smooth muscle cells contractile phenotype [35]; myosin 11, α-actin, transgelin/SM22, calponin-1, myosin regulatory light polypeptide 9, and topomyosin β chain (Appendix A, yellow dots) between vehicle and CAG-treated rats.

## 4. Discussion

In the present study, we aimed to test the proposed protective effects of CAG supplementation on AAA progression. We found that daily administration of CAG significantly attenuated expansion of intraluminal elastase-induced AAA in rats. The aneurysms displayed more preserved elastic lamellae. The underlying mechanism could be linked to diminished aneurysmal MMP-2 activity.

Preservation of elastic lamellae in the CAG treated AAAs could be explained by decreased degradation of elastin or augmented synthesis. Our data suggest that it may not be caused by increased elastin synthesis, as LOX mRNA levels were unchanged. However, we did detect a non-significant upregulation of fibulin-5 by our explorative proteome analysis (FC 1.15 and unadjusted *p*-value = 0.020). Both LOX and fibulin-5 enable the formation of elastin fibers in the aorta by binding to structural proteins such as tropoelastin and fibrillin-1, thereby facilitating increased elastin assembly in AAAs [8]. Moreover, in cultured rat vascular smooth muscle cells, CAG restored the TNF-mediated reduction in expression of fibulin-5 and -1 [19]. The effect of CAG is, however, more likely caused by decreased MMP-2 activity in the CAG-treated aneurysms produced in vascular smooth muscle cells. This is in line with previous murine studies showing a reduced AAA expansion associated with decreased MMP activity, both in the murine elastase wrapping model and the angiotensin II-induced AAA model in hyperlipidemic ApoE^−/−^ mice [19]. That CAG directly affects vascular smooth muscle cells and, thereby inhibits MMP activity, has been shown in TNF-stimulated cultured primary rat vascular smooth muscle cells; the affected molecular signaling pathway was ascribed to dampening of the ERK/JNK signaling pathway [19].

One of the critical elements in AAA progression includes chronic inflammation associated with continuous infiltration of macrophages and lymphocytes because of degradation of the ECM in the aneurysm wall. The infiltrated immune cells release pro-inflammatory cytokines and proteases that activate VSMCs to phenotypic shift and increased collagen production as a compensatory mechanism for the degradation of elastin fibers. Eventually VSMCs become stressed and undergo apoptosis resulting in high production of RNS and ROS [36,37,38,39]. In murine models, inflammation is the primary driver of AAA progression in the first two weeks of AAA expansion [40,41]; thus, this could explain why we, in this study, did not observe any effect of CAG on the monocyte/macrophage marker F4/80 at the mRNA level, along with no reduction of CD68 positive macrophages in the wall of the CAG treated aneurysms. A reduction in infiltrating CD68 positive cells in the aneurysm wall has previously been observed after CAG treatment in aneurysms in mice induced by local elastase wrapping around the abdominal aorta, and in mice where the component 3,4-benzopyrene (an active ingredient in cigarettes) enhanced angiotensin II AAA two weeks, but also six weeks, after AAA induction [19,42]. The underlying mechanisms were ascribed to reduction in transforming growth factor beta (TGFβ) and nuclear factor ĸB-induced production of pro-inflammatory cytokines resulting in diminished inflammation. A similar effect of CAG on macrophages was observed in chronic psoriatic skin lesions in mice, where CAG administration reduced infiltration of macrophages and decreased mRNA levels of the pro-inflammatory cytokines IL-1β, TNF, and IL-6 in the inflamed skin and in LPS stimulated bone marrow-derived macrophages in vitro. The effect of CAG was shown to dampen inflammation by preventing NRLP3 inflammasome activation in bone marrow derived macrophages [18].

As there is consensus that CAG seems to inhibit macrophage infiltration and expression of pro-inflammatory cytokines, we would have expected to detect lower mRNA expression levels of IL-6 in the CAG-treated aneurysms. IL-6 is believed to act as a chemoattractant for immune cells in aneurysms. However, it has not been proven to individually affect progression or expansion of AAAs in mice [43]. Our findings of no difference in IL-6 expression between groups contradicts the existing literature [19]. IL-6 belongs to one of the early inflammatory response genes in inflamed tissue [44]. However, we examined the aneurysmal tissue after 28 days, a phase that, in rats, is more regenerative [40]; macrophages shift to favor the anti-inflammatory and tissue repairing phenotype [12,45], which could be why we did not detect any difference in IL-6 levels.

The balance between M1 and M2 macrophages is important for tissue homeostasis [12]. Dale et al. demonstrated that favoring M1 macrophages augmented AAA expansion, while favoring M2 macrophages dampened AAA progression [9]. Thus, we expected that CAG administration would favor the M2 phenotype. We did not find any difference between groups in our semi-quantifications of CD206 positive cells in aneurysm wall cross-sections. In addition, the mRNA levels of the anti-inflammatory cytokine IL-10 in the aneurysms were not significantly different, nor were mRNA levels of iNOS used as an indicator of M1 macrophages in the aneurysms. Thus, in our study, CAG did not seem to favor a shift from M1 to M2 macrophages. No change in inflammatory status was observed in the aneurysm wall. This corresponds with the absence of changes in the media layer of α-actin positive cells and the fact that we did not detect any difference in quantitative proteins associated with vascular smooth muscle cell contractile phenotype.

As previously described, ROS contribute significantly to AAA progression in both human and murine models [46,47] as ROS promote macrophages to release pro-inflammatory cytokines such as IL-6. Astragaloside IV (AST) obtained from *Astragalus membranaceus* is easily converted by intestinal microbes to CAG by deglycosylation [48]. In murine models of AAA, both AST and CAG reduced ROS and, thereby, dampened the release of pro-inflammatory cytokines IL-6, TNF and MMPs from macrophages [19,42]. The underlying mechanism relates to augmented Nrf2 and HO-1 signaling pathways. Nrf2 is a transcription factor that controls the expression of antioxidant genes [10]. Thus, upregulation of Nrf2 will diminish ROS and, thereby, ROS-mediated inflammation [46]. HMOX1 is a cellular stress response gene regulated by Nrf2 that produces HO-1. HO-1 is responsible for the oxidative cleavage of heme groups released from damaged erythrocytes in the vascular wall, leading to the generation of biliverdin with antioxidant properties, thereby scavenging ROS, carbon monoxide with vasodilatory properties, and release of ferrous iron. Thus, HO-1 has important antioxidant, anti-inflammatory, and cytoprotective effects in vascular cells. Thus, the presence of HO-1 protects vascular smooth muscle cells and endothelial cells from further damage in response to injury. Furthermore, HO-1 deficiency in mice augments AAA progression [49]. Moreover, in humans, polymorphisms in the promoter region of the HMOX1 gene, resulting in decreased expression of HO-1, are associated with increased risk of developing AAA [50]. In our experiments, neither Nrf2 nor HO-1 mRNA levels changed after CAG treatment, suggesting that, in our setting, CAG did not influence ROS or exert antioxidative effects at the examined time point. Perhaps, this could be explained by species difference or the dose of CAG used. The studies showing that CAG or AST reduced the expression of Nrf2 and HO-1 were performed in mice using two different models: the elastase wrapping model and the angiotensin II and 3,4 benzopyrene-induced AAA model in 8–10 months old C57BL/6 mice. As in our experiments, CAG was given orally. In the angiotensin II and 3.4-benzopyrene model, they used daily doses of 20 mg/kg and 80 mg/kg for 6 weeks [42], while CAG, in the elastase wrapping model, was given in a low dose 62.5 mg/kg or high dose 125 mg/kg daily for 14 or 28 days perorally, starting at the day of experiment or at 14 days (high dose only) after AAA induction [19]. We initiated our CAG administration the day after surgery to prevent pre-priming of the aortic wall prior to elastase treatment. The high dose in our experiment was chosen based on the daily dose of CAG on an oral no-observed-adverse-effect-level (NOAEL) > 150 mg/day in rats, achieved by oral administration of 150 mg CAG/kg/day for 91 consecutive days [51] and corresponded to the dose used in Wang et al. [19], though there might be species difference.

To find new potential mechanisms that CAG might affect to dampen AAA progression, we used an explorative proteome approach. We did not find any deregulated proteins in the aneurysm wall in comparing the two groups, when adjustment for multiple testing was done. This might reflect the highly heterogenous tissue that requires numerous samples to detect differences. Amongst the unadjusted regulated genes with a *p*-value below 0.02 was prostacyclin synthase (PGIS) (FC 1.27, unadjusted *p*-value = 0.006). PGIS produces prostacyclin with known vasodilatory, anti-inflammatory, and anti-thrombotic properties counteracting the prothrombotic thromboxane [34,52,53]. The water channel AQP1 was also upregulated (FC 1.26, unadjusted *p*-value = 0.01). We have previously shown that loss of AQP1 accelerates angiotensin II-induced atherosclerosis in hyperlipidemic mice [29]. The underlying mechanism was not identified; however, AQP1 channels in the endothelial cells could, perhaps, contribute to washing out substances, such as LDL, trapped in the subendothelial intimal layer in areas of endothelial dysfunction.

The protective effect of CAG could also be mediated by lowering arterial blood pressure. We did not measure blood pressure in our study, but we did observe a difference in heart to body weight ratios, with a higher ratio among CAG-treated rats that could suggest an elevated blood pressure. However, we believe that the elevated heart to body ratio more likely relates to the relatively lower body weights within the CAG-treated rats, as liver and spleen to body ratios were also slightly elevated. Others have reported that the compound astragaloside IV, which is converted by intestinal microbes to CAG [54], did not affect arterial blood pressure in pregnant rats [55]. They did, however, observe a dose-dependent (20–80 mg/kg) blood pressure lowering effect in preeclampsia-induced pregnant rats, as well as a reduction in preeclampsia-induced oxidative stress [55], suggesting that CAG in our model could potentially have a minor blood pressure lowering effect rather than elevating blood pressure. In comparison, we treated our rats with a higher dose of CAG that seemed to be well tolerated; the rats had a similar weight gain as the vehicle treated controls, in line with the existing literature [19,51].

There are some limitations to the study. Our primary end point was progressive aneurysm dilatation. Therefore, we chose the PPE AAA model in rats, as it is, to our knowledge, the model that presents most of the features of the human disease [20,40,41]. All models are short term, while human AAA develops over years [1,6]. The length of the experimental protocol is, in most studies, either 14 days or 28 days. The first 14 days of AAA expansion is fastest and involves inflammation and oxidative stress as drivers [56]. While, in the last 14 days, AAA expansion declines and reaches a plateau, reflecting tissue-repairing mechanisms with extensive elastin production. To get the full effect of CAG on AAA expansion, we chose to end the experiment after 28 days, when AAA size and elastin integrity could be analyzed, while proinflammatory responses may be less pronounced. Although the reduction in MMP-2 activity seen at day 28 in the aneurysm wall may reflect reduced inflammation and/or oxidative stress at an earlier timepoint.

The effect of CAG in already established AAA has yet to be evaluated. This would be clinically relevant. Although, in our experiments, CAG treatment was provided after induction of AAA by elastase infusion. Thus, we did not affect the aortic wall by initiating CAG treatment prior to AAA induction, which suggests that CAG could likely be beneficial in existing AAA and is supported by findings in mice [19].

## 5. Conclusions

In conclusion, CAG reduced experimental AAA progression. Our data suggest that underlying mechanisms might be mediated by reduced MMP-2 activity and by preserving elastin and reduced calcification. Based on these findings, CAG should be considered as a possible candidate for future dietary supplementation that may dampen AAA expansion in humans.

## Figures and Tables

**Figure 1 biomedicines-10-00359-f001:**
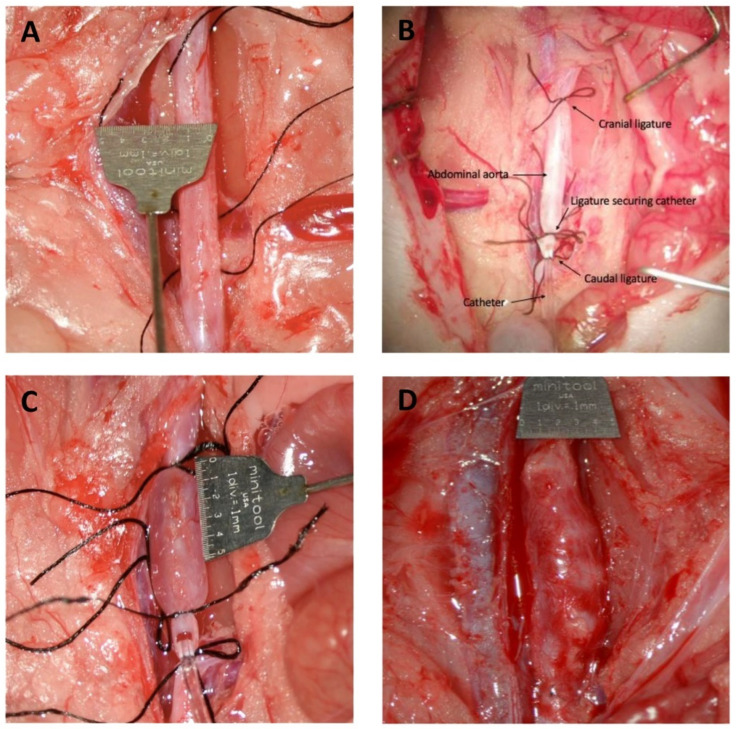
Induction of the surgical steps. (**A**) Isolated infrarenal aorta, (**B**) placing catheter in the infrarenal aorta, (**C**) intraluminal porcine pancreatic elastase infusion, (**D**) isolated AAA on day 28 post surgery.

**Figure 2 biomedicines-10-00359-f002:**
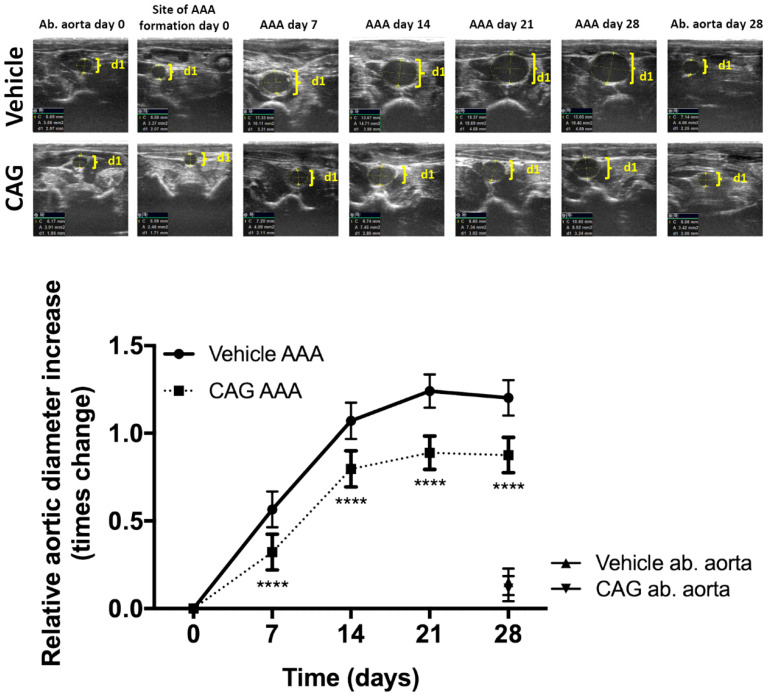
The effect of CAG on AAA growth. Upper panel shows representative ultrasound recordings at baseline and post-surgery days 7, 14, 21, 28 for both vehicle and CAG treated rats. The abdominal aorta (Ab. Aorta) just distal from the left renal artery was used as a reference point for developmental aortic expansion in the experimental period (day 0 and post-surgery day 28). D1: shows aortic diameter. Below, the relative increase in maximal aortic aneurysm diameter adjusted for weight at entry from day 0–28 measured by ultrasound in CAG treated group and vehicle treated group (n = 10/10). Values are mean ± standard deviation. **** indicates *p* < 0.0001.

**Figure 3 biomedicines-10-00359-f003:**
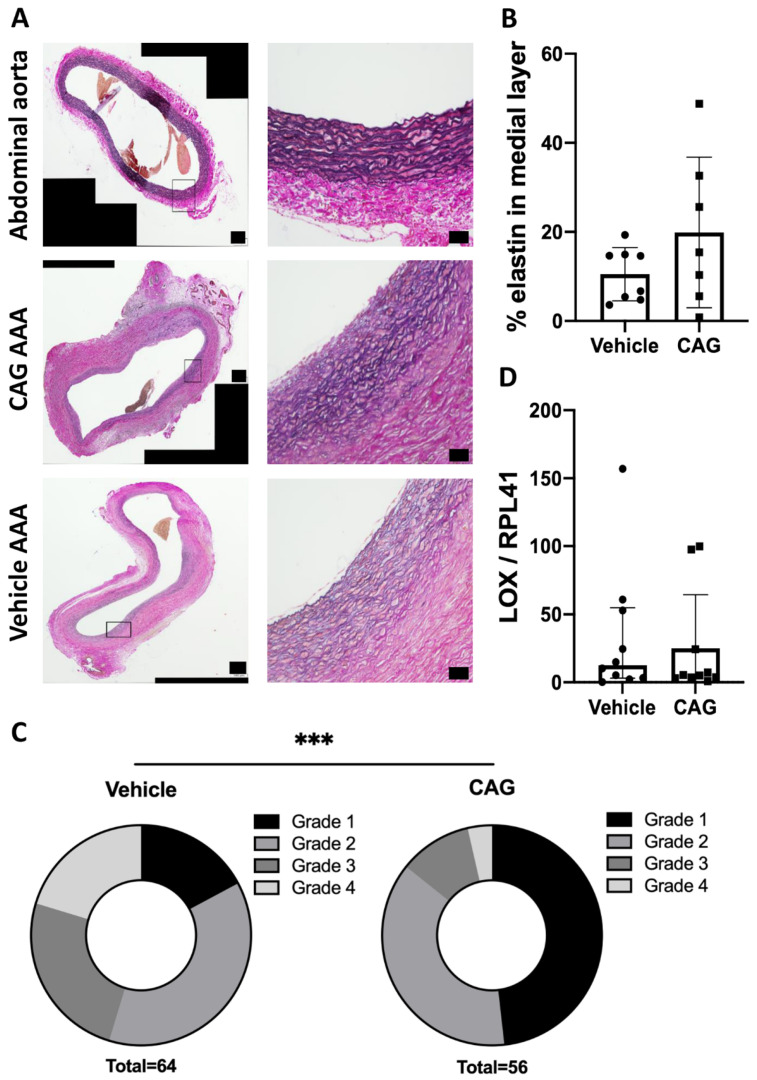
Elastin assessment in abdominal aortic aneurysms (AAA) (**A**) Representative micrographs of Miller’s elastin stain (black) from abdominal aorta, vehicle AAA and CAG AAA at day 28 (enlargements correspond to black square on the left image). Scale bar (black box) in micrographs to the left: 100 μm and micrographs to the right: 50 μm (**B**) Percentage of elastin in medial layer in vehicle and CAG group AAA at day 28 (n = 8/7). (**C**) Scoring of aneurysmal wall elastin disruption in vehicle and CAG group 1–4, 4 being severe wall disruption and 1 minimal wall disruption (n = 8/7; *p* = 0.0003). (**D**) Elastin related mRNA coding for LOX gene (n = 10/10) normalized to RPL41 mRNA levels. Values are median ± inter quartile range. *** Indicates *p* < 0.001.

**Figure 4 biomedicines-10-00359-f004:**
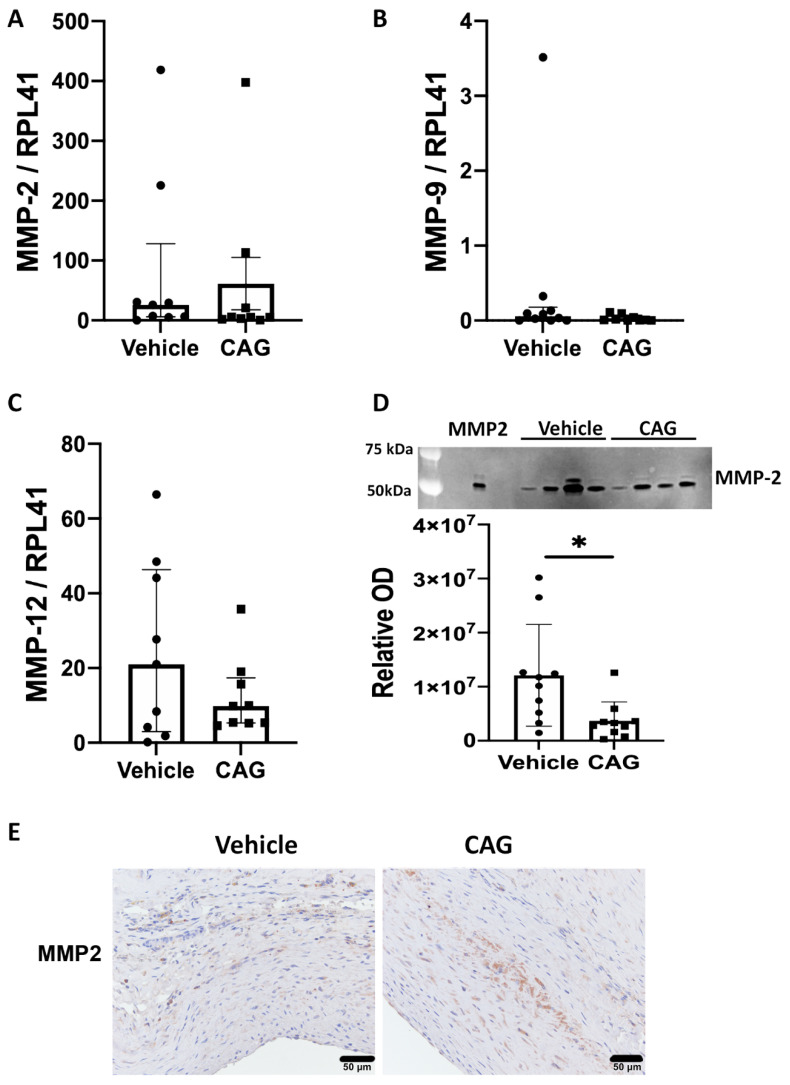
Quantity and activity of matrix metalloproteinases (MMPs) in AAAs. Assessment of messenger RNA (mRNA) from aneurysmal wall samples from vehicle and CAG group coding for (**A**) MMP-2 (n = 9/9), (**B**) MMP-9 (n = 9/9), and (**C**) MMP-12 (n = 9/9). (**D**) Zymography and quantification of zymography showing significantly increased activity of MMP-2 in vehicle compared to CAG group (n = 10/10; *p* = 0.02). (**E**) Displays weak MMP-2 labeling in a subset of vascular smooth muscle cells of the aneurysm wall from both vehicle- and CAG-treated rats (n = 8/7), scalebar = 50 µm. Values in (**A**–**C**) are median ± interquartile range and normalized to RPL41 mRNA levels. Values in (**D**) are mean ± SD, * indicates *p* = 0.02.

**Figure 5 biomedicines-10-00359-f005:**
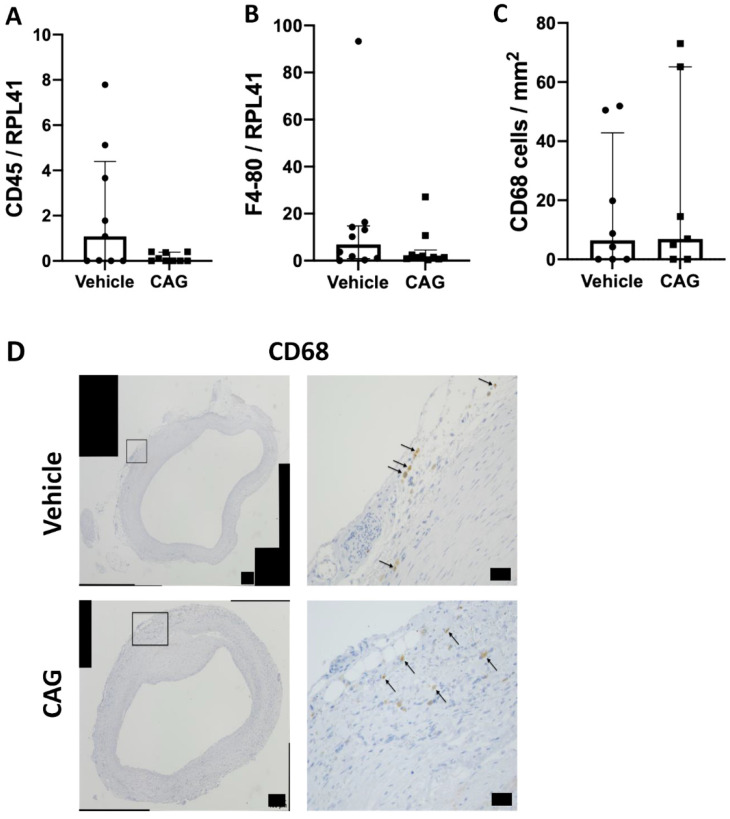
Immune cells in abdominal aortic aneurysms. (**A**) Relative mRNA levels from aneurysmal wall samples from vehicle and CAG treated group of common lymphocyte marker CD45 (n = 9/9), and (**B**) macrophage/monocyte marker F4-80 (n = 10/10). All RNA data were normalized to RPL41 mRNA levels. (**C**) Semi-quantification of aneurysmal CD68 positive cells in AAA of each group per mm^2^ in the adventitial layer (n = 8/7). (**D**) Representative micrographs of CD68 positive cells in vehicle and CAG AAAs on day 28 (enlargement represents black square on the left image). Arrows mark positive cells (n = 8/7). Scale bar (black box) in micrographs to the left: 100 μm and in micrographs to the left: 50 μm. Values are median ± interquartile range.

**Figure 6 biomedicines-10-00359-f006:**
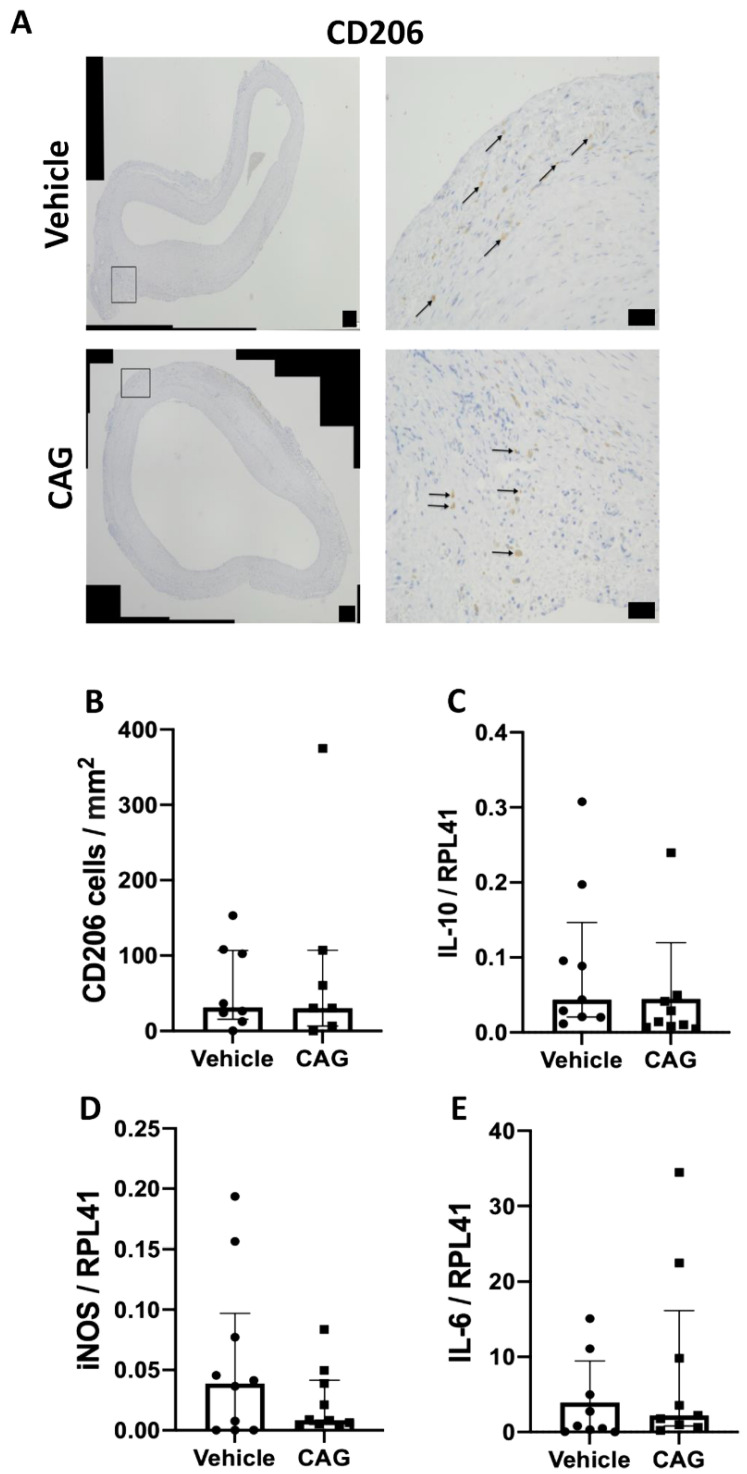
Markers for M1 and M2 macrophages and cytokine expression in the aneurysm wall after 28 days on vehicle or CAG treatment. (**A**) Representative micrographs of CD206 positive staining from vehicle and CAG AAAs at day 28 (enlargement represent black square on the left image). Arrows mark positive cells (n = 8/7). Scale bar (black box) in micrographs to the left: 100 µm and in micrographs to the right: 50 μm. (**B**) Semi-quantification of CD206 positive cells in each group per mm^2^ in adventitia. (**C**) Relative mRNA levels of IL-10 (n = 10/10), (**D**) inducible NO synthase (iNOS) (n = 10/10), and (**E**) IL-6 (n = 9/9) in the aneurysm tissue. All RNA data is normalized to RPL41 mRNA levels. All values are median ± interquartile range.

**Figure 7 biomedicines-10-00359-f007:**
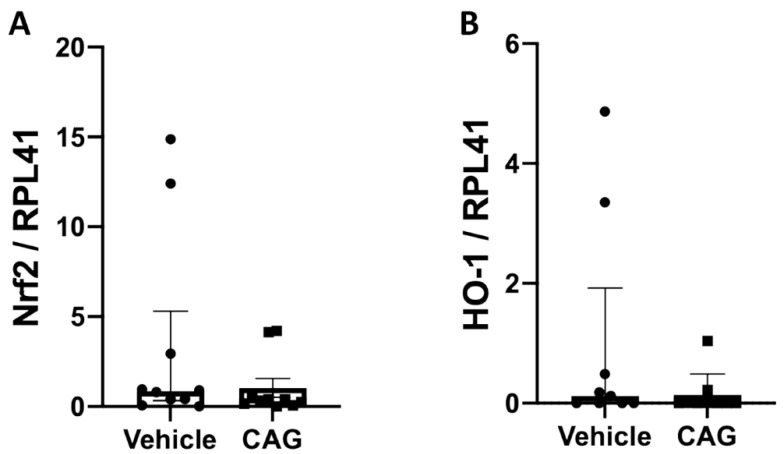
Effect of CAG on aneurysmal oxidative stress. (**A**) Relative mRNA levels of nuclear factor erythroid 2–related factor (Nrf2) (n = 9/9) and (**B**) relative mRNA levels of Heme oxygenase (HO)-1 (n = 9/9) in the aneurysm tissue. All values are median ± interquartile range. All RNA data is normalized to RPL41 mRNA levels.

**Figure 8 biomedicines-10-00359-f008:**
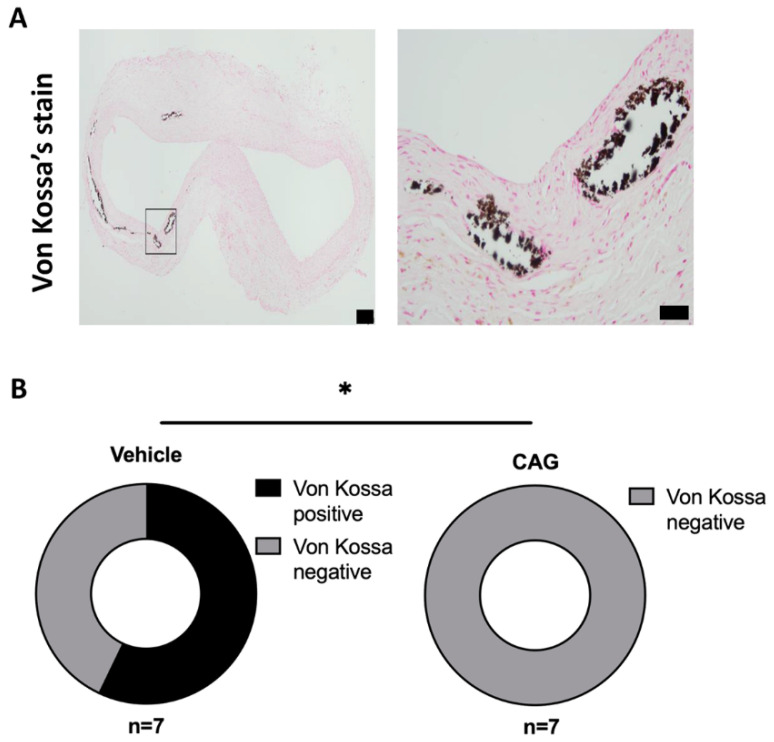
The effect of CAG treatment on aneurysm calcification. (**A**) Representative micrographs of calcium deposits (Black) in the aneurysm wall visualized by Von Kossa’s staining (enlargement represent black square on the left image). Scale bar (black box) in micrograph to the left: 100 μm and in micrograph to the right: 50 μm. (n = 7/7). (**B**) Donut plot of percentage of Von Kossa’s positive aneurysm sections in CAG and vehicle treated groups (n = 7/7; *p* = 0.018). All values are median ± interquartile range. All RNA data is normalized to RPL41 mRNA levels. * Indicates *p* < 0.05.

**Figure 9 biomedicines-10-00359-f009:**
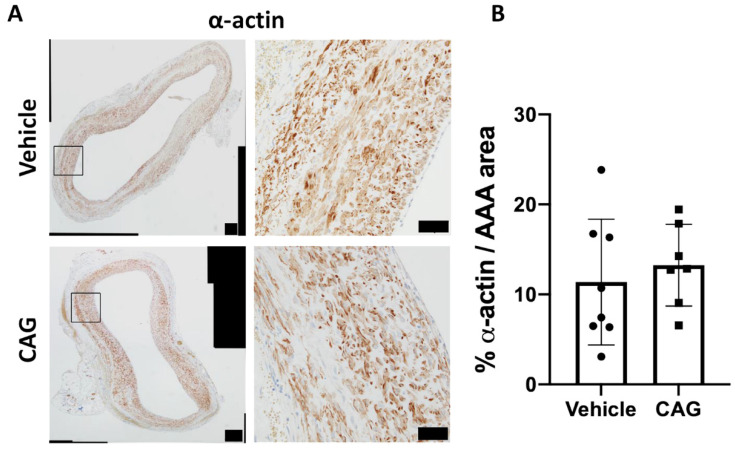
CAG does not affect α-actin positive area in the aneurysm wall. (**A**) Representative micrographs of α-actin staining from vehicle and CAG AAAs at day 28 (enlargement represent black square on the left image, (n = 8/7). Scale bar (black box) in micrographs to the left: 100 µm and in micrographs to the right: 50 μm. (**B**) Semi-quantification of α-actin positive area of total AAA area. Values are mean ± SD.

**Table 1 biomedicines-10-00359-t001:** Primer sequences for qPCR analyses. The coefficient of correlation obtained for the standard curve expressed as R^2^-value is stated for each PCR product.

Target	Forward Primer (5′-3′)	Reverse Primer (5′-3′)	R^2^-Value
Lysyl oxidase (LOX)	ACCTGGTACCCGATCCCTAC	AGTCTCTGACATCCGCCCTA	0.99
Inducible nitric oxidase synthase (iNOS)	AGGCAAGCCCTCACCTACTT	GATGGGAACTCTTCCAGCAC	0.98
Mature macrophages (F4/80)	TTTTGGCTGCTCCTCTTCTG	TGGCATAAGCTGGACAAGTG	0.98
Interleukin-6 (IL-6)	CAGAGTCATTCAGAGCAATAC	CTTTCAAGATGACTTGGATGG	0.98
Interleukin-10 (IL-10)	TCTCCCCTGTGAGAATAAAA	TAGACACCTTTGTCTTGGAG	0.96
Matrix Metalloprotease 2 (MMP-2)	GATCTTCTTCCTTCAAGGATCG	TACACGGCATCAATCTTTTC	0.99
Matrix Metalloprotease 9 (MMP-9)	TACTTTGGAAACGCAAATGG	GTGTAGGATTCTACTGGG	0.99
Matrix Metalloprotease 12 (MMP-12)	CAATATTGGAGGTACGATGTG	GTCATATTCCAATTGGTAGGC	0.90
Cluster of differentation 45 (CD45)	GCTATAAAAGACCCCTTCAG	CATTAGGCAAATAGAGACACTG	0.99
Heme oxygenase 1 (HO-1)	ACAGAAGAGGCTAAGACCG	CAGGCATCTCCTTCCATT	0.99
Nuclear factor erythroid-2-related factor (Nrf2)	CCATTTGTAGATGACCATGAG	CTATTAAGACACTGTAACTCGG	0.95
Ribsomal Protein L41 (RPL41)	TGGCGGAAGAAGAGAATGC	TGGACCTCTGCCTCATCTTT	0.99

**Table 2 biomedicines-10-00359-t002:** Top Deregulated Proteins in CAG Treated AAA vs. Vehicle Treated AAA.

Accession	Description	Fold Change	*p*-Value
O08658	Nuclear pore complex protein	1.17	0.001
Q9Z1X1	Extended synaptotagmin-1	1.15	0.003
P61227	Ras-related protein Rap-2b	0.77	0.003
P20171	GTPase HRas OS = Rattus norvegicus	0.79	0.004
P53534	Glycogen phosphorylase, brain form (Fragment)	1.14	0.004
P21263	Nestin	1.71	0.005
Q62969	Prostacyclin synthase	1.25	0.007
O35353	Guanine nucleotide-binding protein subunit beta-4	1.22	0.010
Q4V8H8	EH domain-containing protein 2	1.16	0.012
P09414	Nuclear factor 1 A-type	1.39	0.014
O89043	DNA polymerase alpha subunit B	1.19	0.014
Q8CF97	Deubiquitinating protein VCIP135	0.80	0.014
P63029	Translationally-controlled tumor protein	0.90	0.014
P29975	Aquaporin-1	1.25	0.016
Q62745	CD81 antigen	1.07	0.016
B2RYW9	Fumarylacetoacetate hydrolase domain-containing protein 2	1.20	0.018
Q7TQ16	Cytochrome b-c1 complex subunit 8	1.18	0.018
P60892	Ribose-phosphate pyrophosphokinase 1	1.14	0.019
Q9JLZ1	Glutaredoxin-3	0.93	0.020
Q9WVH8	Fibulin-5	1.15	0.020

## Data Availability

The mass spectrometry proteomics data have been deposited to the ProteomeXchange Consortium via the PRIDE [57] partner repository with the dataset identifier PXD030441.

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
