# Peer review of "Cycloastragenol Inhibits Experimental Abdominal Aortic Aneurysm Progression"

_biomedicines, 2022, doi:10.3390/biomedicines10020359_

Round 1

Reviewer 1 Report

The study by Melin et al. describes the potential protective effect of CAG treatment to abdominal aortic aneurysm development. The authors present significant reduction of AAA diameter that could be explained by lower elastin degradation and calcification. However, no changes in infiltration of inflammatory cells or matrix metaloproteinses activity was noticed. The study is interesting, but it’s lacking some important points. Below is the list of my major concerns.

The authors could provide the representative USG records in the M or B-mode to illustrate the AAA development in time.

The authors should explain why they used RPL41 as a reference gene. Its expression differs in tissues therefore it may not be the best choice for normalization of gene expression. Did the authors analyze other reference genes such as EF2 or CyP A?

The authors should consider doing in situ zymography of analyzed matrix metalloproteinases on aortic scraps as it was described e.g., by Kopacz et al. (doi.org/10.1155/2020/6340190). It could show the localization of active MMPs and could explain the bias results of Figure 4D.

The authors should verify the protein level of antioxidative gene – HO-1 and transcriptional factor Nrf2. The gene expression of these two give little information on their activity. Especially, Nrf2 should be analysed at the protein level using e.g., immunofluorescence to verify if the CAG activates Nrf2 and if it translocates to the nucleus.

Why the authors claim that CAG treatment influence MMP2 level when it was not changed? Also, why the authors combine on one figure antioxidative genes and calcification (Figure 7)?

Basing on the presented results the conclusions are a bit exaggerated and should be adjusted to gathered results.

Author Response

Comments and Suggestions for Authors

1. The study by Melin et al. describes the potential protective effect of CAG treatment to abdominal aortic aneurysm development. The authors present significant reduction of AAA diameter that could be explained by lower elastin degradation and calcification. However, no changes in infiltration of inflammatory cells or matrix metaloproteinses activity was noticed. The study is interesting, but it’s lacking some important points. Below is the list of my major concerns.

ANSWER: Thank you for the summary of our results. It is correct we did not observe any change in aneurysm MMP2 mRNA levels between vehicle and CAG treated rats. We did however observe a decrease in aneurysm MMP2 activity, determined by zymography. We agree that in our data the measured level of MMP2 was unaffected, but the MMP2 activity was reduced in the CAG treated group.

2. The authors could provide the representative USG records in the M or B-mode to illustrate the AAA development in time.

ANSWER: Thank you for this great suggestion, we have now included representative USG recordings for all time points including the start and endpoints of abdominal aorta just distal from the left renal artery in figure 2.”The effect of CAG on AAA growth. Upper panel shows representative ultrasound recordings at baseline and post-surgery days 7, 14, 21, 28 for both vehicle and CAG treated rats. The abdominal aorta (Ab. Aorta) just distal from the left renal artery was used as a reference point for developmental aortic expansion in the experimental period (day 0 and post-surgery day 28). D1: shows aortic diameter.”

3. The authors should explain why they used RPL41 as a reference gene. Its expression differs in tissues therefore it may not be the best choice for normalization of gene expression. Did the authors analyze other reference genes such as EF2 or CyP A?

ANSWER: Thank you for suggesting alternative reference genes. Many genes work well as reference genes. We always test if the chosen reference gene used, changes by the experimental condition. In this case, we found no significant difference in aneurysm RPL41 mRNA levels between vehicle and CAG-treated animals.  We, therefore, used RPL41 as a reference gene in this study. We have included in the method section, page 8, “Each mRNA expression level of the gene of interest was normalized to the complimentary expression level of the housekeeping gene Ribosomal Protein L41 (RPL41), that we first tested and did not change significantly between vehicle and CAG treated aneurysms”

4. The authors should consider doing in situ zymography of analyzed matrix metalloproteinases on aortic scraps as it was described e.g., by Kopacz et al. (doi.org/10.1155/2020/6340190). It could show the localization of active MMPs and could explain the bias results of Figure 4D.

ANSWER: Yes, we agree that in situ zymography would provide the localization of active MMPs in the aneurysm wall and thereby would provide additive information of where in the aneurysm wall protease activity are present. We chose to use a quantitative method by measuring protease activity by zymography rather than a more descriptive method, though both analyses would be optimal.

We are unfortunately unable to perform in situ zymography on the current material as we have either fixed tissue samples in formalin for histology or used the tissue for quantitative measurements. We have stained the aneurysm sections for MMP2 immunoreactivity.

Page 13MMP2 was associated with a subset of vascular smooth muscle cells in the aneurysm wall, where weak labeling of MMP2 was detected in the medial layer. There was no apparent difference between the two groups (Figure 4E).”

5. The authors should verify the protein level of antioxidative gene – HO-1 and transcriptional factor Nrf2. The gene expression of these two give little information on their activity. Especially, Nrf2 should be analyzed at the protein level using e.g., immunofluorescence to verify if the CAG activates Nrf2 and if it translocates to the nucleus.

ANSWER: Yes, that would be interesting to examine. However, we do not have the antibody in-house, and to make the revision within the given 10 days. we were unable to add these experiments to the data set. Because the time to receive the antibody, test it, and make the analyses was too short, for this given revision. We will take it into consideration for future analyses.

To compensate for the lack of available antibody, we have tried to identify HO-1 and Nrf2 in our explorative proteomic analyses, however both HO-1 and Nrf-2 was below detection levels of this assay.

6. Why the authors claim that CAG treatment influence MMP2 level when it was not changed? Also, why the authors combine on one figure antioxidative genes and calcification (Figure 7)?

ANSWER: We conclude that CAG treatment affects MMP2 activity. We base this on our zymography data. We have now moderated it a bit to: in the abstract: "Cycloastragenol inhibits experimental abdominal aortic aneurysm progression. The suggested underlying mechanisms involve decreased matrix metalloprotease-2 activity, and preservation of elastin and reduced calcification." 

We also agree it may be confusing to combine the effect of CAG treatment on oxidative genes and aneurysm calcification. We have now divided figure 7 into two separate figures. (Figure 7and Figure 8).

 7. Basing on the presented results the conclusions are a bit exaggerated and should be adjusted to gathered results.

ANSWER: Thank you for your comment. We have in our abstract adjusted our conclusion, page 1, line 28: “Cycloastragenol inhibits experimental abdominal aortic aneurysm progression. The suggested underlying mechanisms involve decreased matrix metalloprotease-2 activity and preservation of elastin and reduced calcification.”

As well, as the conclusion at the end of the manuscript has been adjusted, page 19, line 28 “CAG reduces experimental AAA progression. Our data suggest that underlying mechanisms might be mediated by reduced MMP2 activity MMP-2 activity and preservation of elastin and reduced calcification.”

Reviewer 2 Report

In this study, authors investigated the protective function of cycloastragenol (CAG) to the development of abdominal aortic aneurysm (AAA). CAG treatment significantly attenuated PPE-induced AAA enlargement in rats. Authors investigated the possible roles of CAG on anti-inflammatory and anti-oxidative effects. However, all the major inflammatory markers and oxidative markers tested, did not showed different between CAG treated and control rats and only MMP-2 activity and arterial calcification were significantly attenuated in the CAG-treated aortas when compared with the vehicle aortas.

Major concerns are:  

  1. Rats at the age of 7-10 weeks are at young adult phase during development (PMID: 23930179), the baseline growth and remodeling of the aorta during the selected age range need to be included in the current study. The actual aortic diameters at different timepoints are preferred to show the in vivo phenotype, as shown in the paper PMID: 32857766. Alternatively, a later adult age would be better to show the aortic phenotype.
  2. A possible false-positive finding in the proteomics assay because the adjusted analysis did not show any significant changes while unadjusted analysis was performed despite a small cut-off p-value of 0.02. The total number of identified proteins was not shown in the manuscript, which is important to evaluate a false positive. Thus, the highlighted proteins do not provide convincing evidence to prove the hypothesis, and other unidentified mechanisms are involved in the CAG-attenuated AAA phenotype under PPE induction.
  3. Since levels of inflammatory markers and oxidative markers tested did not showed different between CAG treated and control rats, authors should consider to investigate whether CAG treatment alters endothelial cell or smooth muscle cell function, such as TGF-beta signaling.
  4. MMP-2 activity was significantly decreased in the CAG treated rat group. Authors should perform enzymatic analysis to test whether CAG can directly and specifically inhibitor MMP-2 activity.

Minor comments:  

  1. Quantification analyses in histology study should be standardized: elastic fiber breakage or fragmentation number is proved to be a solid and objective aortic medial disruption parameter, which is much better than the subjective 4-grade classification analyses.  
  2. The sample number should be consistent in all the figures, histology samples should be 8 and 7 while the fresh samples (mRNA or protein) should be 10 and 10 in the vehicle and CAG groups, respectively.  

Author Response

In this study, authors investigated the protective function of cycloastragenol (CAG) to the development of abdominal aortic aneurysm (AAA). CAG treatment significantly attenuated PPE-induced AAA enlargement in rats. Authors investigated the possible roles of CAG on anti-inflammatory and anti-oxidative effects. However, all the major inflammatory markers and oxidative markers tested, did not showed different between CAG treated and control rats and only MMP-2 activity and arterial calcification were significantly attenuated in the CAG-treated aortas when compared with the vehicle aortas.

Major concerns are:  

  1. Rats at the age of 7-10 weeks are at young adult phase during development (PMID: 23930179), the baseline growth and remodeling of the aorta during the selected age range need to be included in the current study. The actual aortic diameters at different timepoints are preferred to show the in vivo phenotype, as shown in the paper PMID: 32857766. Alternatively, a later adult age would be better to show the aortic phenotype.

ANSWER: Thank you for your comments. As described in the suggested paper (PMID:23930179) Rats live up to 2.5-3.5 years, and such old rats are not commercially available and would be difficult to handle due to their size. Most important are that they are sexually matured (week 7, even though they are not mentally matured). They produce testosterone from week 7, which is needed for AAA development.

We have ordered our rats based on the body weight of the rats, which corresponds to young adult rats. There was a large variation in the initial body weight of the rats and thus also the initial diameter of the abdominal aorta (1.57-2.01 mm). We, therefore, used diameter increase as readout and adjusted statistical analyses tor initial body weight. The increase of abdominal aorta just distal of the renal branch has now been included in figure 2 to show that the increase seen in the aneurysms is not because of rats growing during the experimental period. We have included representative ultrasound recordings from both treatment groups in figure 2.

  1. A possible false-positive finding in the proteomics assay because the adjusted analysis did not show any significant changes while unadjusted analysis was performed despite a small cut-off p-value of 0.02. The total number of identified proteins was not shown in the manuscript, which is important to evaluate a false positive. Thus, the highlighted proteins do not provide convincing evidence to prove the hypothesis, and other unidentified mechanisms are involved in the CAG-attenuated AAA phenotype under PPE induction.

ANSWER: Thank you for your comment. We agree that the cut-off p<0.02 is very arbitrary and that there might be some false-positive findings. So, we have changed the cut-off to the top 20 de-regulated proteins in the aneurysm wall.  We have now included on pages 20-21“We identified 2011 unique proteins (minimum n = 3/3), of which 57% were detected across all samples (n = 10/10) (Supplementary Table 1). No significant differences were found between CAG treated and vehicle treated aneurysms when correcting for multiple testing (Supplementary Figure 2), so one should bear in mind that some unadjusted de-regulated proteins might be false positive."

The top 20 de-regulated proteins in the aneurysm wall are shown in Table 2.” by our explorative proteome analyses. All raw data are available at ProteomXchange Consortium via the PRIDE partner repository with the dataset identifier PXD030411.

  1. Since levels of inflammatory markers and oxidative markers tested did not showed different between CAG treated and control rats, authors should consider to investigate whether CAG treatment alters endothelial cell or smooth muscle cell function, such as TGF-beta signaling.

ANSWER: Thank you for this interesting suggestion. We have now semi-quantified whether CAG affects the vascular smooth muscle cell layer in the aneurysm cross-sections identified by alpha-actin staining. Here we did not see any difference in the alpha-actin positive area. Also, we have included findings from our explorative proteome analysis and identified proteins associated with vascular smooth muscle cell contractile phenotype (myosin 11, actin, transgelin/SM22, calponin-1, myosin regulatory light polypeptide 9 and topomyosin beta-chain) they are all included in supplementary table 2 and shown as yellow dots in supplemental figure 2. These proteins were not affected by CAG -treatment.

    We have added a new paragraph in the results sections page 21:

3.6 Effect of CAG on vascular smooth muscle cells.

To determine if vascular smooth muscle cell layer in the aneurysm wall were changed by CAG treatment the α-actin positive cells were examined in aneurysm cross-sections from the two groups. Intense α-actin labeling was detected in the medial layer of the aneurysms in both groups and there was no difference in the area of positive α-actin staining between vehicle and CAG treated rats (Figure 9 A, B, p= 0.56). That no major change in vascular smooth muscle cell layer was observed was further supported by the quantitative proteome analyses of the aneurysm wall showing no change in proteins associated with vascular smooth muscle cells contractile phenotype [35]; myosin 11, α-actin, transgelin/SM22, calponin-1, myosin regulatory light polypeptide 9, and topomyosin β chain (supplementary table 2 and supplementary figure 2, yellow dots) between vehicle and CAG treated rats.“                  

And in the discussion page 23: “No change in inflammatory status was observed in the aneurysm wall. This corresponds with no changes in the media layer of α-actin positive cells and that we did not detect any difference in quantitative proteins associated with vascular smooth muscle cell contractile phenotype.”

Due to the time limitation for revising this manuscript (10 days), we were unable to perform additional experiments to examine the role of TGFbeta signaling. By going through our proteome data, we found that proteins related to TGFbeta signaling: Transforming growth factor beta-1-induced transcript 1 protein (FC= 1.10, unadjusted p-value= 0.142, n=10/10), TGF-beta receptor type-1 (FC=0.94, unadjusted p-value= 0.59, n=6/7), TGF-beta receptor type-2 (FC=1.43, unadjusted p-value=0.15, n=3/3), TGFbeta1 proprotein (FC:0.91, unadjusted p-value=0.47, n=3/3). All detected proteins were not changed in the CAG treated group. Even though not all TGF related proteins were detected in all samples, we find it unlikely that these protein levels would be changed by CAG treatment.

We did manage to run a single western blot testing aneurysm protein samples from vehicle and CAG treated rats of the downstream target of TGF-beta signaling by phosphorylated Smad-2 (at site ser465/467, Cell Signaling) and normalize it to β-actin. There was no significant difference of p-Smad2 between vehicle and CAG treated rats (1.55 ± 0.37 vs 1.07± 0.11, n=4/5, p=0.286 by unpaired Students t-test with Welch’s correction). Since we only manage to test half of the aneurysm protein samples, we felt these data were too preliminary to be included in the manuscript.

  1. MMP-2 activity was significantly decreased in the CAG treated rat group. Authors should perform enzymatic analysis to test whether CAG can directly and specifically inhibitor MMP-2 activity.

ANSWER: Yes, we agree it would be interesting to confirm that CAG directly can inhibit MMP2 activity. This has already been shown in vitro in primary vascular smooth muscle cells from rats where CAG inhibited the TNF-induced MMP2 activity seen in their mouse AAA models (PMID:3030274). As we were given only 10 days to revise the manuscript, we were unable to confirm these findings because of limited time.

We have now included I the discussion page 21: “That CAG directly affects vascular smooth muscle cells and thereby inhibiting MMP activity has been shown in TNF stimulated cultured primary rat vascular smooth muscle cells, and the affected molecular signaling pathway was ascribed to dampening of the ERK/JNK signaling pathway [19]

Minor comments:  

  1. Quantification analyses in histology study should be standardized: elastic fiber breakage or fragmentation number is proved to be a solid and objective aortic medial disruption parameter, which is much better than the subjective 4-grade classification analyses.  

ANSWER: Thank you for the suggestion of an alternative parameter for elastin disruption assessment.
In our 4-grade classification analyses, our investigators included the density of each elastic lamellae, but also fiber breakage and general wall disruption. Based on these factors, each area was scored.

We agree that quantitative counting of elastin fiber breakage and fragmentation number would be a relevant supplement to the grading system. Unfortunately, due to limited time to revise the manuscript of 10 days, we have not been able to add these additional analyses to our study.

  1. The sample number should be consistent in all the figures, histology samples should be 8 and 7 while the fresh samples (mRNA or protein) should be 10 and 10 in the vehicle and CAG groups, respectively.  

 ANSWER: Thank you for the comment. We have in the manuscript accounted for the differences in sample size between analyses. We have under termination section 2.7 explained that samples used for histological analyses were damaged on page 5, subsection 2.7, “Two samples from the vehicle-group and three from the CAG-group were unfortunately damaged in the embedding process and were therefore not included in the histological analyses (n=8/7). "

Regarding Von Kossa’s staining, we have stated on page 5, subsection 2.8: “One additional sample from the vehicle group was damaged during the Von Kossa staining process. (n=7/7)."

For RNA analyses as stated on page 7, subsection 2.13, line 41: “RNA yield in one sample from each group was low; therefore, these samples were only included in RPL41, LOX, F4/80 and iNOS”. This means that n=10/10 for LOX-, F4/80- and iNOS relative mRNA levels. For the relative mRNA levels of IL6, IL10, MMP2, MMP9, MMP12, CD45, HO-1 and Nrf2 (n=9/9).

Round 2

Reviewer 1 Report

I am content with the authors corrections and answers to my questions.

Reviewer 2 Report

Authors have addressed my concerns.

This manuscript is a resubmission of an earlier submission. The following is a list of the peer review reports and author responses from that submission.